# Nuclear Magnetic Resonance Fingerprints and Mini DNA Markers for the Authentication of Cinnamon Species Ingredients Used in Food and Natural Health Products

**DOI:** 10.3390/plants13060841

**Published:** 2024-03-14

**Authors:** Subramanyam Ragupathy, Arunachalam Thirugnanasambandam, Varathan Vinayagam, Steven G. Newmaster

**Affiliations:** Natural Health Products (NHP) Research Alliance, College of Biological Sciences, University of Guelph, Guelph, ON N1G 2W1, Canada; vinaychemist89@gmail.com (V.V.); snewmast@uoguelph.ca (S.G.N.)

**Keywords:** cinnamon, genome skimming, NMR fingerprinting, herbs, spices, authentication

## Abstract

*Cinnamomum verum* (syn *C. zeylanicum*) is considered ‘true’ cinnamon. However, it is reported that less expensive sources of cinnamon from *C. cassia* (syn *C. aromaticum*), *C. loureiroi*, and *C. burmannii* (toxic coumarin) may be used in the place of *C. verum*. We lack the quality assurance tools that are required to differentiate *C. verum* from other cinnamon species when verifying that the correct species is sourced from ingredient suppliers. The current research on cinnamon species authentication using DNA tools is limited to a few species and the use of high-quality DNA extracted from raw leaf materials. The cinnamon bark traded in the supply chain contains much less DNA and poorer-quality DNA than leaves. Our research advances DNA methods to authenticate cinnamon, as we utilized full-length chloroplast genomes via a genome skimming approach for *C. burmannii* and *C. cassia* to facilitate the design of optimal mini DNA markers. Furthermore, we developed and validated the use of NMR fingerprints for several commercial cinnamon species, including the quantification of 16 molecules. NMR fingerprints provided additional data that were useful for quality assessment in cinnamon extract powders and product consistency. Both the new mini DNA markers and NMR fingerprints were tested on commercial cinnamon products.

## 1. Introduction

Cinnamon is one of the most popular flavoring agents in the food and beverage industry. Although it is mostly used in the food and spice industry as a flavor ingredient, there are some medicinal plant studies that suggest that it has anti-diabetic activity [1,2]. Cinnamon is the common name used for several species in the genus *Cinnamomum*. The dried bark of *Cinnamomum verum* J. Presl (syn *C. zeylanicum* Blume; Ceylon Cinnamon) is considered ‘true’ cinnamon, which is native to Sri Lanka, but is also grown and sourced from other countries in Africa, South America, and the West Indies [3]. It has been reported that *C. cassia* (L.) J. Presl (syn *C. aromaticum* Nees; Chinese cassia), *C. burmannii* (Nees & T. Nees) Blume (Indonesian cassia), and *C. loureiroi* Nees (Saigon cassia) are used in the place of *C. verum* in cinnamon in Europe and North America [4,5]. Cinnamon has several health benefits, including its use as a blood thinner and treatment for several types of cancer, diabetes, and Alzheimer’s [6,7,8,9,10]. Cinnamaldehyde is the major compound found in all the *Cinnamomum* species [11] and has been used as an antibacterial agent and to treat diabetes [11,12,13]. However, coumarin—a potentially toxic compound—is present in Cassia cinnamon, which is sometimes used as a substitute for true cinnamon in food and herbal products. Although coumarins have a role in human health [14], the US Food and Drug Administration banned the use of coumarin in food products due to its ability to produce hepatotoxicity in animal models at high doses [15,16]. It is therefore necessary to differentiate *C. verum* from other commercial cinnamon species.

Cinnamon phytochemistry can be used for quality control using a variety of different analytical technologies. Different species of *Cinnamomum* have unique chemical components in varying amounts based on species, processing, and geography [17,18,19,20]. These species-specific differences in phytochemistry result in the higher value of *C. verum* compared to *C. cassia*, which is a common source in the cinnamon supply chain [4]. Consequently, the need for the supply chain verification of species authenticity has resulted in the development of analytical methods [21], including the quantification of coumarin as required by regulatory agencies [22,23]. HPTLC was developed for the quality control of cinnamon but has several limitations regarding accurate cinnamon species identification; it requires considerable resources to develop methods for each type of product form, it lacks a universal detector, co-elution is considerable, and this method requires highly qualified technicians [23,24,25,26,27]. Mass spectroscopy (MS) methods have been developed to quantify specific components found in several species of commercial cinnamon [28,29]. The use of (DART)-QtoF-MS was developed to discriminate ‘true cinnamon’ (*C. verum*) from several cinnamon species (*C. burmannii*, *C. cassia*, and *C. loureiroi*) based on chemical composition [4]. More recently, simpler methods that are not as costly, such as NIR hyperspectral imaging (NIR-HSI) and FT-NIR/FT-IR spectroscopic techniques have been developed for cinnamon species (*C. burmannii*, *C. cassia*, and *C. loureiroi*) authentication [5,30,31,32]. These methods detect several compounds and employ chemometrics to identify different species of *Cinnamomum*. Further advances have resulted in the use of a rapid method, based on X-ray fluorescence, of identifying the major compounds unique to Ceylon cinnamon from other adulterants, using multivariate analysis [33]. The methods above are based on targeted analytical chemistry with some assumptions regarding the presence of targeted bioactive compounds found in specific species.

Metabolomics using nuclear magnetic resonance (NMR) provides both targeted and untargeted metabolite profiles from a test sample, including the structure and quantification of specific compounds. NMR for metabolite characterization and fingerprinting is a modern approach that is routinely used for the investigation of the quality control of pharmaceutical drugs [34], food [35,36,37,38], and herbal products [39,40]. Specific NMR methods have been developed for cinnamon products, such as oils [41] and ground powders [42], including the quantification of compounds such as coumarin, which can be toxic at higher concentrations [20]. This is particularly useful in dealing with mixtures of different species or complex compounds without the need for any chemical separation step using another technique [43]. Research has demonstrated that NMR methods can be used to identify the source (twig, bark) and geographic origin of a specific cinnamon sample [44]. Farag et al. [45] were the first to develop an NMR method for the authentication of cinnamon products with *C. verum* and *C. cassia* species. They later validated this method on commercial products and identified several metabolites ((E)-methoxy cinnamaldehyde and coumarin) to differentiate *C. verum* and *C. cassia* [19]. Further research is needed to build NMR spectral fingerprint libraries for more commercial cinnamon species, including those from different biological materials and geographic origins. Furthermore, the studies to date include key uncertainties concerning the validated authenticity of the samples, as they were secured from commercial markets. The samples used to build NMR fingerprint libraries need to be authenticated and labelled with their known provenance.

Several DNA-based tools have been developed as molecular diagnostic tools to identify cinnamon taxa. An early study by Kojoma et al. [46] used the *trnL* intron and *trnL*-*trnF* intergenic spacer to successfully differentiate *C. burmannii*, *C. cassia*, *C. sieboldii*, and *C. verum*. Universal and degenerate ITS primers designed to identify medicinal plants [47] were used by Lee et al. [48] to differentiate *C. burmannii*, *C. cassia*, *C. loureiroi*, and *C. verum*. Although they were not specifically designed for studying *Cinnamomum* spp., these primers were able to distinguish the above-mentioned species using DNA from leaves. However, both studies used good-quality and -quantity DNA from leaves and primer pairs that yielded 193 bp (ITS2) and 429 bp (*trnL*-*trnF*) amplicons for species identification. DNA barcoding was proposed as an authentication method based on the use of several DNA regions [48,49,50,51]. Methods for DNA isolation have been developed specifically for *Cinnamomum* species based on leaves [52,53,54]. Although the further development of DNA barcode tools has included both fresh leaf and ground bark samples [55,56,57], these methods can still yield ambiguous results for some samples/species [58]. In a previous project, we tested these primers on commercial samples comprised of dried cinnamon bark powders and were not able to retrieve the target sequences. This was not surprising because (1) the DNA in these processed samples is broken into fragments that are too short for the primers designed for much longer amplicon sizes [59], and (2) DNA fragments are degraded, damaged, and sheared in these processed samples [60]. Fit-for-purpose tools need to consider the form of the test materials, and, in the case of the processed materials used in the industry, these molecular diagnostic tools need to recover short DNA fragments.

The use of short DNA mini barcodes and probes requires a phylogenetic approach using a larger sampling of the cinnamon genome. Having full-length chloroplast genomes can provide extra sequence information for the development of multiple assays based on short amplicons. Previous studies showed that the whole chloroplast genome provides sequence information for approximately 100 genes [61]. Recently, Bandaranayake et al. [62] published all the data for several species of *Cinnamomum*, resolving genetic relationships, including the differentiation of populations of *C. verum* from India and Sri Lanka. This research was underpinned by several fully sequenced *Cinnamomum* chloroplast genomes, including a few samples from the commercial species *C. verum*, *C. burmannii*, and *C. cassia* [63,64,65,66,67,68,69,70]. The focus of these whole-genome studies was on phylogeny, including many non-commercial species. There are key uncertainties concerning specific regions in the genome that can be used to differentiate commercial species, including no primers for mini DNA markers that could be used to authenticate commercial cinnamon products. Further studies are needed to confirm the whole-genome sequence of key commercial species to reveal smaller regions that can differentiate cinnamon species.

The goal of this study is to develop orthogonal methods for the authentication and quality assessment of several commercial cinnamon species based on species reference materials. More specifically, we focused on developing two methods:Development of mini DNA markers—we utilized genome skimming to retrieve plastid regions through the shallow sequencing of two commercial cinnamon species: *C. burmannii* and *C. cassia*. Then, we assembled full-length chloroplast genomes for *C. burmannii* and *C. cassia* from high-throughput sequence data using the chloroplast genome of *C. verum* as a reference. The goal was to provide broader genome sampling that could be used in a phylogenetic approach to differentiate cinnamon species and further the development of mini DNA markers. These markers could then be properly validated [71] and used in on-site qPCR platforms for quick, affordable, quality assurance tools.Development of NMR Fingerprints—commercial samples, with their species identity verified using DNA methods, were used to develop NMR spectral fingerprints on a 400 MHz Bruker AVANCE III. The goal was to develop quick screening methods and more detailed analysis of cinnamon spectra for use in quality control systems that could verify species identity from raw to processed samples throughout the supply chain. We utilized NMR spectra for each species to quantify molecules of interest to provide further assessment of quality assurance and possible efficacy for health claims.

## 2. Results

### 2.1. Chloroplast Genome

#### 2.1.1. Chloroplast Genome Assembly

We generated a total of 7,916,620, and 16,648,994 paired-end reads for *C. burmannii* (FC165), and *C. cassia* (379NW), respectively (Figure 1). The de novo assembly generated 643 contigs with an N50 length of 3146 bp and a total length of 2.3 Mb for *C. burmannii* (FC165), and 8372 contigs with an N50 length of 2762 bp and a total length of 24.2 Mb for *C. cassia* (379NW). Draft chloroplast genomes were created for both species after aligning the reads to the reference chloroplast genome of *C. verum* (Figure 1). The complete chloroplast genome of *C. burmannii* was assembled with a total of 152,764 bp in size, and it contained four regions: a large single-copy (LSC) region of 93,704 bp and a small single-copy (SSC) region of 18,911 bp, separated by two inverted repeat regions (IR) of 20,073 bp each. Similarly, the chloroplast genome of *C. cassia* is 152,724 bp long, containing a large single-copy (LSC) region of 93,684 bp and a small single-copy (SSC) region of 18,907 bp, separated by two inverted regions (IR) of 20,065 bp each (Appendix A).

#### 2.1.2. Chloroplast Genome Characteristics

The overall GC content is almost identical for both genomes of *C. burmannii* (39.1%) and *C. cassia* (39.2%). Similarly, the GC content of SSC (33.9%) and two IRs (44.4%) is identical for both species, while LSC has 37.9% and 38% for *C. burmannii* and *C. cassia*, respectively. The chloroplast genomes of *C. burmannii* and *C. cassia* consisted of 123 and 122 different genes and 79 and 78 protein coding genes, respectively. However, both genomes consisted of 36 transfer RNAs (tRNAs) and eight ribosomal RNA genes (rRNAs). In both genomes, the majority of genes were in the large single-copy (LSC) region. In contrast, genes that belonged to the NADH-dehydrogenase group were distributed across the genome, except *ndhC*, *ndhJ*, and *ndhK*, which were present in the LSC region. Introns were identified for 18 genes in both genomes. Except for two genes (*clpP* and *ycf3*), which consisted of two introns, the remaining 16 genes had only one intron. In both genomes, the *trnK*-UUU gene had the largest intron of 2513 bp, which consisted of the *matK* gene, and the smallest intron was found in *trnL*-UAA, which was 479 bp long. The intron length varied from 589 to 1127 bp and from 590 to 1130 bp for *Cinnamomum burmannii* and *C. cassia*, respectively. Intron lengths were found to be consistent between the two species and only varied between 1 and 3 bp in both genomes across seven genes: *atpF*, *clpP*, *ndhA*, *petB*, *rpoC1*, *trnV*-UAC, and *ycf3* (Table 1). Codons coding for the amino acid leucine were the most identified codons in both *C. burmannii* (4090) and *C. cassia* (3968). Similarly, the most abundant codon was ATT (Isoleucine) in both *C. burmannii* (1580) and *C. cassia* (1554). The least abundant codon was TAG (stop codon) in both genomes (2%).

#### 2.1.3. Chloroplast Genome Phylogeny

A chloroplast genome phylogeny for *Cinnamomum* spp. was generated, including the two sequences generated here, along with seventy-one additional chloroplast genome sequences downloaded from Genbank, representing 30 *Cinnamomum* species (Appendix A). *C. verum* and *C. cassia* formed monophyletic clades with samples from other studies, as expected, but the *C. burmannii* samples (including the sample from this study, along with several additional *C. burmannii* sequences from other studies) appeared to be polyphyletic and were not clearly resolved.

#### 2.1.4. Development and Testing of New Mini DNA Markers

The full-length chloroplast genomes of both *C. burmannii* and *C. cassia* (Appendix A, respectively) were aligned with the chloroplast genomes of other *Cinnamomum* species (available in public databases) to identify markers suitable for amplification in processed cinnamon samples that contain sheared DNA. We identified five variable regions in the comparative analysis of the *Cinnamomum* chloroplast genomes. Five primer pairs were designed to amplify short amplicons in the processed botanicals of cinnamon, with their amplicon lengths varying from 130 bp to 290 bp (Appendix A). The five primers were tested on all the samples of the three *Cinnamomum* spp. (*C. burmannii*, *C. cassia*, and *C. verum*). All five primers, Cinbur_129, Cinna_140, Cinbur158, Cincas_216, and Cincas_290, amplified the products across all the samples tested. Sequence analysis of the successful primers identified *C. cassia* and *C. burmannii* samples clearly from *C. verum* samples (Appendix A).

### 2.2. NMR Chemical Fingerprinting

The proton NMR chemical profiles of all 48 samples of three *Cinnamomum* species revealed considerable metabolite diversity among the samples. The binned proton NMR data was hierarchically clustered to determine the similarities and differences in metabolite diversity among all the samples. All species were clearly differentiated and classified into clusters based on similarities in their respective metabolite profiles (Figure 2). Fine-scale clustering was based on variation in the metabolite profiles within species that may be attributed to the plant part (e.g., leaf, stem), geographic origin, or product processing and formulation (Figure 2). The structural elucidation of the molecular structure within the metabolite profiles of each sample enabled the detection of specific molecules. Structural elucidation revealed 16 molecules that characterized approximately 60% of the proton structure found within the NMR spectra of all the samples (Figure 3 and Appendix A). This provided further verification of the specific molecules associated with each species sample (Figure 4).

### 2.3. Quantification of Metabolites

The quantification of 16 metabolites (Appendix A) from 48 different cinnamon samples was completed using nuclear magnetic resonance (NMR) (Table 2). The quantity of metabolites varies depending on the sample type, which includes raw bark, bark powders, powders, and various extracts. The study has identified five significant molecules of interest for analyzing the concentration variation among the samples. These molecules are coumarin, cinnamaldehyde, methoxy cinnamaldehyde, cinnamic acid, and eugenol. Among the samples, the concentration of coumarin is variable among species, with the highest levels of coumarin (9.2 mg/g) found in the *C. cassia* samples. Coumarin has been used to treat primary lymphoedema, but its use is restricted in some countries due to considerable concerns of hepatotoxicity [72]. The European Food Safety Authority has set a safe daily limit for consuming coumarin, called the “tolerable daily intake” (TDI), which is 0.1 mg per kilogram of body weight. The TDI was developed on the basis of there being no observed adverse effects below this level in animals [73]. The TDI limit is intended to prevent individuals from consuming harmful levels of coumarin. One of the samples, labelled “Cin_cass_20”, has a higher concentration of coumarin than the described TDI.

The study also recorded variation in cinnamaldehyde, cinnamaldehyde, cinnamic acid, and Eugenol quantities among the species. Six samples contained higher concentrations, more than double that of the other samples (Table 2). However, the relationship between cinnamaldehyde content and species was not clearly resolved. Methoxycinnamaldehyde quantities were higher in *C. verum* than in the other cinnamon species, with a quantification of almost 6% of the product; this is approximately double the amount of *C. cassia*. Four *C. verum* samples had cinnamic acid quantities approximately four times higher than *C. cassia*. Eugenol is an aromatic flavouring molecule found in cinnamon samples in lower concentrations and is observed in both *C. verum* and *C. cassia* in the range of 1–2%. However, in *C. burmannii*, Eugenol quantities are much lower (0.1–0.6%). Although eugenol has not been associated with liver damage or elevated enzymes, the ingestion of high doses, such as an overdose, could result in severe liver injury [74].

The multivariate statistical analysis of the quantified 16 metabolites has revealed differentiation among the species, as shown in Figure 5. This two-dimensional multivariate classification ordination has explained 71.63% of the variance in the quantified metabolite contents among the different cinnamon species. Moreover, these 16 metabolites could be used to identify species in addition to complete chemical fingerprinting.

## 3. Discussion

### 3.1. Complete Chloroplast Genomes Reveal Important Phylogenetic Relationships in Cinnamon

The phylogenetic classification of *C. burmannii* based on whole-genome sequences was not clearly resolved (Appendix A). *C. verum* and *C. cassia* samples formed monophyletic clades, but the *C. burmannii* samples were found to be polyphyletic. Yang et al. [66] published a whole chloroplast genome sequence for *C. burmannii* and produced a phylogenetic tree including eight other *Cinnamomum* plastid genome sequences. In the study by Yang et al. [66], *C. burmannii* was clearly resolved from the other *Cinnamomum* species, including *C. verum*. However, they only included a single *C. burmannii* accession. We produced a phylogenetic tree based on our whole-genome sequences and 71 other *Cinnamomum* chloroplast genome sequences published in Genbank representing 30 species, including 6 *C. burmannii* plastid genome sequences. The *C. burmannii* sequences appear in clades with several other *Cinnamomum* species (frequently with *Cinnamomum insularimontanum*). Further research is needed to resolve the phylogeny of *Cinnamomum* species, with particular focus on more populations of *C. burmannii*.

### 3.2. Genome Skimming for Mini DNA Marker Development as an Authenticity Screening Tool

DNA barcoding tools have advanced to include more sophisticated genomic methods that consider the broad sampling of the plant genome. The extended barcode concept proposes the use of the shallow-pass shotgun sequencing of genomic DNA to generate sequences for chloroplast and mitochondrial genomes, along with nuclear ribosomal DNA, and shallow coverage of single-copy nuclear DNA [75]. Extended barcodes are suggested as an improvement to overcome the limitations of standard DNA barcodes because they provide additional sequence information for species discrimination from the complete chloroplast genome. In this study, we used a genome skimming approach to generate extended barcodes for testing cinnamon samples. Genome skimming is one of the simplest high-throughput sequencing techniques and involves the random sampling of a small percentage of total genomic DNA (gDNA) through shallow sequencing [61]. Recently, this approach has gained popularity in skimming the genomes of plant samples preserved in museums and herbaria [76,77,78] because of its ability to generate valuable sequence information from degraded and dried material. However, this approach has only been used in a limited number of medicinal plant studies [79,80]. Depending on the plant genome size, type, and age of the plant material used for DNA extraction, genome skimming can generate enough sequence data to assemble full-length chloroplast genomes [81,82]. In the current study, we were able to utilize this novel approach to assemble complete chloroplast genomes for *C. burmannii* and *C. cassia*. We used this new sequence information from full-length chloroplast genomes to design mini DNA markers suitable for testing commercial cinnamon species.

### 3.3. The Current Need to Develop Mini DNA Markers to Test Processed Ingredients and Products

The primary requirement for the authentication of herbal products using DNA-based methods is the availability of good-quality DNA, which is generally obtained from fresh plant material. However, the plant materials present in commercially available botanical products are subjected to storage, drying, grinding, chemical treatments, and extraction methods, resulting in DNA degradation and shearing [59,60,83]. Also, herbal dietary supplements frequently contain other pharmaceutical excipients such as fillers, binders, and stabilizers that can interfere with DNA extraction due to adsorption phenomena [84]. In our study, we found that the DNA quantity was higher in fresh leaves and lower in processed extract powders, which is supported by earlier published research [59,60]. Obtaining adequate DNA quality through optimization and testing different protocols is one approach to improved herbal quality assessment testing using DNA-based methods. Another important strategy is to use DNA markers of appropriate size for amplification in low-integrity DNA [85]. Recently, RAPD (randomly amplified polymorphic DNA) and ISSR (inter-simple sequence repeats) have been used to generate a small 190 bp species-specific SCAR (sequence-characterized amplified region) markers to differentiate *C. verum* from *C. cassia* [86]. The SCAR method was developed in the early 1990s [87] and has been used to authenticate plant species [88]; this method requires the development of RAPD markers and sequence data to design the PCR primers for the SCAR markers [89]. Testing cinnamon botanicals with existing DNA markers of regular barcode length proved to be a challenging task and, in some cases, was not possible for many cinnamon extracts that we tested in our lab. This is due to the low-quality DNA obtained from the processed cinnamon products that were produced from bark that, by nature, has low DNA quantities. Although the markers developed from the *trnL*-*trnF* intergenic spacer by Kojoma et al. [46] differentiated commercially important *Cinnamomum* species using DNA derived from fresh leaf material, our testing indicated that these primers could not amplify the longer 429 bp amplicon in cinnamon powdered bark samples. Similarly, we could not use the existing ITS2 barcode markers on our cinnamon samples, which otherwise proved to be successful in amplifying high-quality cinnamon DNA, due to (1) the degeneracy incorporated into one of their primer sequences [47] and (2) the relatively long amplicon size. Another study [50] used standard barcode markers (*rbcL*, *matK*, and *trnH*-*psbA*) to detect adulteration in traded cinnamon bark samples. They reported that *matK* was not amplified in any of these market samples and *rbcL* was only amplified in 70% of the samples tested, while *trnH*-*psbA* could not differentiate the tested species. Due to these limitations of existing DNA markers to amplify and detect the cinnamon market samples successfully, we used new sequence information from complete chloroplast genomes to differentiate *C. burmannii* and *C. cassia*, which are commonly used for adulterating the commercially valuable *C. verum*. Our comparative analysis of the complete chloroplast genomes of *C. verum* and other *Cinnamomum* spp. in the literature was used to design smaller-size markers suitable for amplification in processed DNA. This provided a foundation for the development of cinnamon mini DNA markers that we successfully used to verify industry samples. We suggest that these mini DNA markers can be further developed into probes and validated for use in the qPCR platforms that are a common quality assurance tool in the food and NHP industries.

### 3.4. NMR Fingerprints Provide an Efficient Multipurpose Quality Assurance Tool

NMR fingerprint analysis provides a different perspective for assessing the quality and authenticity of cinnamon ingredients and finished products. Cinnamon species ingredient authentication using NMR was first demonstrated by Farag et al. [19,45] and was expanded in the research that we present in this paper for more species and applications. DNA approaches have the advantage of offering comparisons with many more non-commercial species if full-length sequences can be recovered. However, cinnamon products are most often produced from bark with little DNA and then processed into powder for use in multiple ingredient matrices in the food and NHP industries. NMR fingerprint standard operating protocols (SOPs) for extracting metabolites are easily developed for these biological materials, which would be extremely difficult or impossible for successful DNA extraction. NMR fingerprints also provide a benchmark for authenticating species ingredients that can be used to ensure that newly sourced samples are not contaminated with other biological ingredients; this is possible using DNA methods but requires considerable resources and is therefore cost-prohibitive for routine quality assurance testing. NMR provides a measure of product consistency that ensures that products are the same from batch to batch. NMR fingerprint consistency has been used in temporal projects to assess the “shelf-life” of a product, i.e., the time it takes for a product/ingredient fingerprint to change, which can be assessed quantitatively, including the presence of specific bioactive molecules [90,91,92]. More recently, in our lab, we have used NMR fingerprint consistency in cinnamon powders and extracts at different points in time, providing a quantitative measure of biomolecule change to assess the shelf-life or freshness of cinnamon products. NMR fingerprints can be used to assess fine-scale variation in metabolite profiles associated with geographic location, agronomy, processing, or other factors that might influence plant biochemistry production (on the farm) or degradation (oxidation, etc.). This has been demonstrated in many studies, including for coffee [93,94], cocoa [95,96], olive oil [97,98], wine [99], and other food commodities [100]. Although we recorded fine-scale variation in the NMR spectra within species, we did not have enough data to classify the likely source of the variation (geographic origin, etc.). However, we are assembling a larger NMR fingerprint library for cinnamon and other commodities that will include the fine-scale variation that will be useful to verify supply sources and the quality of cinnamon. NMR spectra have the inherent molecular structure of molecules present in a sample and, therefore, we were able to identify specific molecules, which is crucial to differentiate between different species of cinnamon. These cinnamon species-specific molecules are currently used in targeted analytical methods such as HPTLC, MS, and FT-NIR/FT-IR spectroscopic techniques to verify the quality of ingredients in the supply chain [23,27,101]. In our study, we identified 16 bioactive molecules found in cinnamon and quantified them to verify their quality, as needed for label requirements, health claims, and species authentication or to provide an organoleptic cinnamon model for flavor, aroma, etc. This is very efficient, as one NMR sample test can provide a multipurpose metabolite fingerprint for quality assessment, providing solutions concerning ingredient authenticity, consistency, origin, flavor, aroma, etc. There are considerable applications of NMR for ingredient organoleptic quality such as flavor profiles related to metabolite profiles. This has been demonstrated and is now utilized in the industry for many commercial foods and spices [102,103,104]. Changes in cinnamon NMR spectra may be used to predict quality and flavor profiles through time for high-quality brands that seek to provide consumers with the best culinary experience. To facilitate the identification of cinnamon species and bioactive molecules, we have built a chemotaxonomic library for use in quality assessment. We suggest that this library is useful for the industry when assessing an unknown sample, as the spectra in our library can be used to verify the cinnamon species and bioactive molecules present in a supplier’s sample. This information is important for the quality control and authentication of cinnamon products.

## 4. Materials and Methods

### 4.1. DNA Samples Extraction and Sequencing

In total, 22 samples of three species were gathered, including leaf and bark samples (15) and industry powdered samples of known provenance (7) from the Natural Health Products Research Alliance (NHPRA) collections, College of Biological Sciences, University of Guelph, Canada (Appendix A). Sample collection protocol complies with national, international, and institutional guidelines. The thirteen leaf samples of all species were identified using morphological (floristic and vegetative) characteristics and traditional taxonomic methods. The list of samples included seven *Cinnamomum burmannii* samples, five *C. cassia* samples, and ten *C. verum* samples. Herbarium vouchers were created for each specimen and deposited in the College of Biological Science (CBS), Natural Health Products Research Alliance (NHPRA) OAC Herbarium at the University of Guelph.

DNA was extracted for all samples using the guidelines mentioned in DNeasy Plant Mini Kit (Qiagen, Hilden, Germany). Extracted DNA samples were sent to Genomics Facility of the Advanced Analysis Centre (AAC) at the University of Guelph, and DNA libraries were prepared using Nextera DNA Flex kit (Illumina, San Diego, CA, USA). Genome skimming was performed on an Illumina MiSeq platform at the AAC (University of Guelph).

PCR for the newly developed mini primers was performed under standard conditions, as described in Fazekas et al. [105]. PCR products were bidirectionally sequenced using BigDye™ sequencing reactions, and sequence products were analyzed on an ABI 3730 DNA Analyzer (Thermo Fisher Scientific, Waltham, MA, USA) at the University of Guelph AAC Genomics Facility. Chromatographic traces were edited and assembled into contiguous alignments using CodonCode Aligner (version 10.0.2).

### 4.2. Assembly and Annotation of Chloroplast Genomes

We used CLC Genomics Workbench (v 21.0.3) for assembling the chloroplast genome from raw data. Paired-end reads were mapped randomly against the reference chloroplast genome (*Cinnamomum verum*) using the ‘Map Reads to Reference’ tool available on the CLC workbench. The parameters performed in CLC are as follows: match score of 1, mismatch cost of 2, deletion and insertion costs of 3, length fraction of 0.5, and similarity fraction of 0.8. After aligning the reads, consensus chloroplast genomes were generated. Draft chloroplast genomes were created after refining the consensus sequences with re-mapping of the reads. Assembled chloroplast genomes were annotated using CPGAVAS 2 [106] web server. We used default parameters for annotation of the genomes and selected *Cinnamomum camphora* as a reference for annotation; it has clear annotations available on GenBank.

A full chloroplast genome phylogeny was built for *Cinnamomum* spp. using the two chloroplast genomes (*C. burmannii* and *C. cassia*) generated for this study, combined with all *Cinnamomum* complete chloroplast genomes available in Genbank (excluding sequences marked as UNVERIFIED). *Laurus nobilis* was used as the outgroup. Sequences were aligned using MAFFT [107] with auto settings. Some of the Genbank sequences were found to have started at non-standard locations on the chloroplast genome and were adjusted to start at the beginning of the LSC (near *trnH*-GUG) to facilitate alignment. Some of the sequences also had the SSC region in the inverted orientation, and these were adjusted to allow for proper alignment (the orientation of the SSC region can vary within species/individuals, and this is generally not phylogenetically informative [108]). A maximum likelihood analysis was performed using RAxML [109] with 1000 bootstrap replicates.

### 4.3. Nuclear Magnetic Resonance (NMR) Sample Preparation Methods

In total, 48 samples of 3 species were used from the NHPRA collections, including 7 powdered samples used for DNA analysis in this study (Appendix A); note that the bark and leaf samples were not used in the NMR study as they have different chemical profiles to powdered samples, whereas the DNA in all sample matrices contains the same DNA sequence profiles. The methods of extraction and NMR analysis were refined from NMR methods developed in earlier publications by Martinez-Farina et al. [40], Kesanakurti et al. [79], Burton et al. [38], and Kim [110]. For NMR spectroscopy, all 48 samples were extracted in triplicate. Then, 300 mg of homogenized samples were weighed and placed into a 15 mL glass vial, then a solution of methanol-d4 (200 mL) and methanol (1800 mL) containing 0.05% (*w*/*w*) TMS (Tetramethylsilane) was added to the vial. The sample vials were tightly closed with a polypropylene screw cap and vortexed for 2 min, sonicated in a bath at 40 Hz for 5 min, and centrifuged for 15 min at 6000 rpm at room temperature, and then the clear supernatant (650 μL) was transferred to a 5 mm NMR tube for ^1^H-NMR spectral acquisition.

### 4.4. Nuclear Magnetic Resonance (NMR) Spectroscopy

This study utilized 1-dimensional proton NMR NOESY spectra collected on the 400 MHz Bruker AVANCE III in deuterated liquids. To ensure high-quality spectra, 64 scans were performed for each sample. The data were analyzed using Bruker Topspin 4.0.7, with 32 K data point resolution. MestReNova (version 14.0.0) was used for peak assignment and identification of organic molecules. The spectra for 48 *Cinnamomum* samples were acquired based on the resonance signals from the nature of protons of metabolites present. Variations among the NMR signal positions and amplitudes indicate the metabolite content and concentration variation among the 48 samples. The ^1^H-NMR spectra were processed using TopSpin 4.0.7, where phase and baseline were checked and corrected to ensure high-quality spectra. The spectra were calibrated to the TMS/TMSP peak at 0.0 ppm. The quantification of identified metabolites was completed using the Bruker ERETIC2 [111] method with precision and accuracy supported using two steps. Firstly, in the calibration process, the Bruker QuantRefA sample was utilized as the ERETIC2 reference to ensure a reliable outcome. In this quantification step, the NMR spectra of all cinnamon samples were meticulously analyzed. Secondly, to ensure the highest level of precision, the 1D 1H NMR quantification of each sample was performed in triplicate. Processed spectra were bucketed with simple rectangular buckets of positive intensities without scaling (AMIX 4.0.1), with a chemical range from 1 to 12 ppm and a width of 0.01 ppm. During bucketing, the residual solvent signals of water, methanol, and TMSP were removed at the regions from 4.75 to 5.06, from 3.16 to 3.45, and from −0.05 to 0.05 ppm, respectively. After bucketing, each spectrum was normalized by setting the below means as 0, and the above means were binned from 1 to 100. Multivariate statistical methods, as described by Martinez-Farina et al. [38] and Kesanakurti et al. [79], were used to analyze NMR spectra and classify species based on the similarity of metabolite fingerprint spectra. The structural elucidation/confirmation of bioactive compounds in the *Cinnamomum* spp. was completed using 1D NOESY and 2D COSY/TOCSY experiments. The library of reference bioactive compounds at NHPRA is assigned individually and spiked with the samples. Any issues with chemical shifts in the spectra were rectified using the spiking studies and knowledge of the compound structures [112].

## 5. Conclusions

Today’s food industry presents considerable challenges for managers of ingredient/product quality. Sophisticated processing methods challenge existing methods that are used to verify species ingredient authenticity. The modern supply is extremely complex, with ingredients that are sourced from global suppliers that must be vetted via efficient screening tests. In this study, we present novel orthogonal methods for the quality assurance of cinnamon ingredients and products. Cinnamon mini DNA markers can be used as a quick screening test with simple platforms such as PCR instruments that are common to most industry labs. Cinnamon NMR fingerprints are multipurpose, providing detailed information on species authenticity/adulteration, consistency, and bioactive molecule quantification that can be used to identify the supplier source and quality metrics such as flavor profiles. Quantification estimates of the major molecules among the different cinnamon species provide a basis for further mechanistic studies, including clinical trials focused on the efficacy of cinnamon species-specific health benefits and toxicity concerns. This paper is an example of a research method for one species that could be modelled for more species to advance more rigorous quality assessment methods and a movement towards supporting a dose-based approach to the bioactive molecules found in NHPs.

## Figures and Tables

**Figure 1 plants-13-00841-f001:**
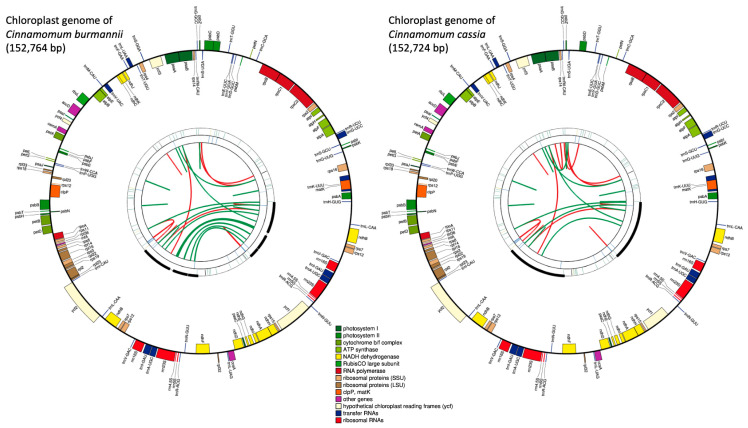
Chloroplast genome maps of *C. burmannii* and *C. cassia* generated via CPGAVAS2. Each map contains four rings. From the center going outward, the first circle shows the forward and reverse repeats connected with red and green arcs, respectively. The second circle shows the tandem repeats marked with short bars. The third circle shows the microsatellite sequences identified using MISA. The fourth circle shows the gene structure on the plastome. The genes are colored based on their functional categories, which are shown at the bottom.

**Figure 2 plants-13-00841-f002:**
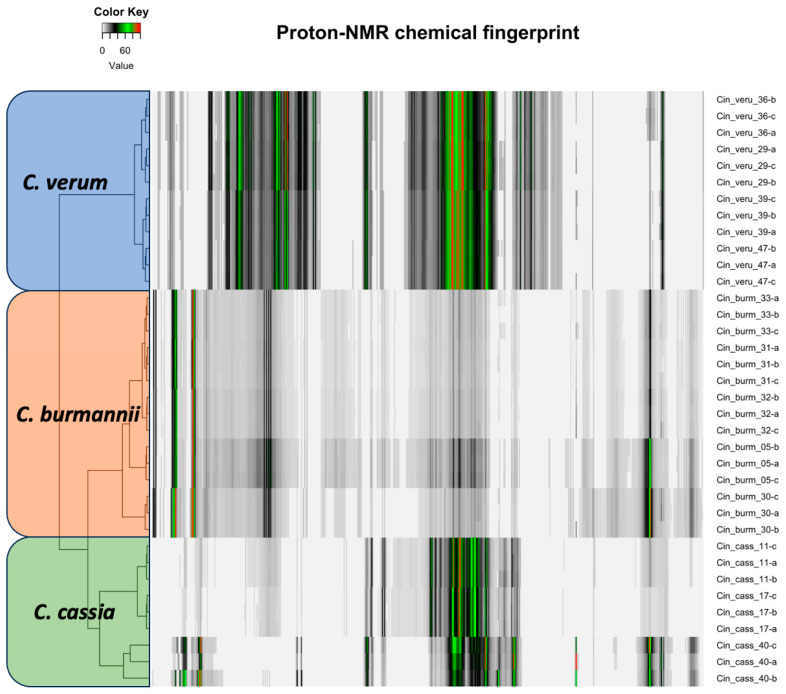
The proton NMR profile of three cinnamon species of the same plant part (root powder). Each bar here represents the peak and intensity of the proton NMR spectra and is hierarchically clustered according to species.

**Figure 3 plants-13-00841-f003:**
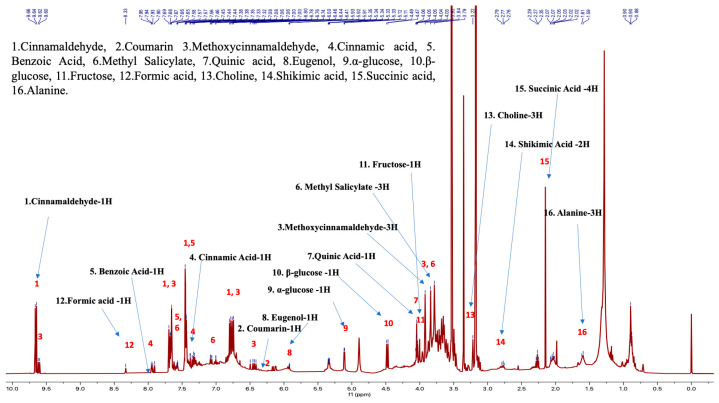
Cinnamon NMR profile with the structural elucidation and assignment of 16 bioactive molecules in the chemotaxonomic library (sample proton NMR spectra of *C. verum*).

**Figure 4 plants-13-00841-f004:**
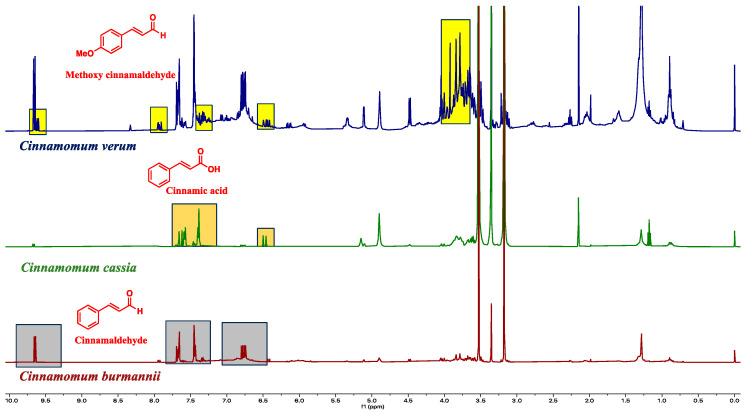
The species-specific molecules are assigned in the proton NMR spectra. The rations of the active molecules are used for species authentication.

**Figure 5 plants-13-00841-f005:**
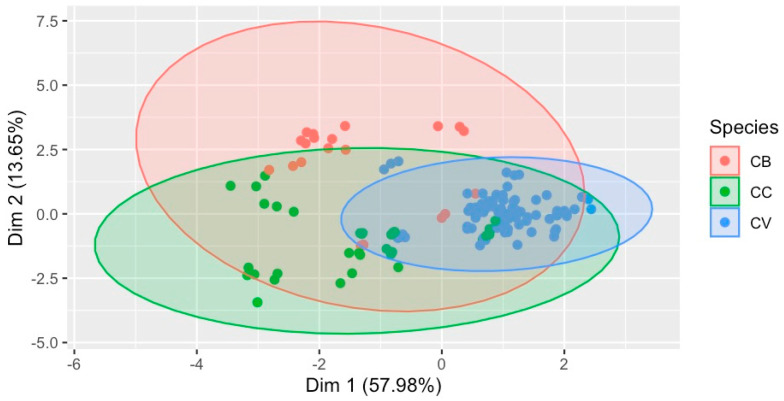
Classification of species based on the metabolite (16) contents in the sample, with 71.63% variance covered. Species: CB—*C. burmannii*, CC—*C. cassia*, and CV–*C. verum*.

**Table 1 plants-13-00841-t001:** Exon/intron regions of *C. burmannii* and *C. cassia*.

***Cinnamomum burmanii* Exon/Intron Locations**								
**Gene**	**Strand**	**Start**	**End**	**ExonI**	**IntronI**	**ExonII**	**IntronII**	**ExonIII**
trnK-UUU	−	1786	4370	37	2513	35		
rps16	−	5193	6306	40	844	230		
trnG-UCC	+	10,328	11,149	23	751	48		
atpF	−	13,053	14,333	145	726	410		
rpoC1	−	21,858	24,650	453	720	1620		
ycf3	−	44,714	46,685	124	734	230	731	153
trnL-UAA	+	49,428	49,991	35	479	50		
trnV-UAC	−	54,470	55,132	39	589	35		
clpP	−	73,555	75,592	71	773	294	656	244
petB	+	78,470	79,905	6	788	642		
petD	+	80,103	81,301	8	716	475		
rpl16	−	84,735	86,110	9	971	396		
rpl2	−	87,837	89,328	392	670	430		
ndhB	−	97,853	100,033	721	702	758		
trnI-GAU	+	105,605	106,620	37	944	35		
trnA-UGC	+	106,685	107,555	38	798	35		
ndhA	−	124,323	126,541	553	1127	539		
trnA-UGC	−	138,914	139,784	38	798	35		
trnI-GAU	−	139,849	140,864	37	944	35		
ndhB	+	146,436	148,616	721	702	758		
***Cinnamomum cassia* Exon/Intron Locations**								
**Gene**	**Strand**	**Start**	**End**	**ExonI**	**IntronI**	**ExonII**	**IntronII**	**ExonIII**
trnK-UUU	−	1789	4373	37	2513	35		
rps16	−	5189	6302	40	844	230		
trnG-UCC	+	10,317	11,138	23	751	48		
atpF	−	13,042	14,324	145	728	410		
rpoC1	−	21,849	24,642	453	721	1620		
ycf3	−	44,706	46,677	124	735	230	730	153
trnL-UAA	+	49,419	49,982	35	479	50		
trnV-UAC	−	54,461	55,124	39	590	35		
clpP	−	73,541	75,576	71	772	294	655	244
petB	+	78,454	79,888	6	787	642		
petD	+	80,086	81,284	8	716	475		
rpl16	−	84,720	86,094	9	970	396		
rpl2	−	87,818	89,309	392	670	430		
ndhB	−	97,833	100,013	721	702	758		
trnI-GAU	+	105,577	106,592	37	944	35		
trnA-UGC	+	106,657	107,527	38	798	35		
ndhA	−	124,295	126,516	553	1130	539		
trnA-UGC	−	138,882	139,752	38	798	35		
trnI-GAU	−	139,817	140,832	37	944	35		
ndhB	+	146,396	148,576	721	702	758		

**Table 2 plants-13-00841-t002:** Quantification of 16 identified metabolites using the ERETIC2 method in mg/g. Average (M) and standard deviations (SD) are provided.

Sample_ID	Cinnamaldehyde		Coumarin		Methoxy Cinnamaldehyde		Cinnamic Acid		Benzoic Acid		Methyl Salicylate		Quinic Acid		Eugenol		α-Glucose		β-Glucose		Fructose		Formic Acid		Choline		Shikimic ACID		Succinic Acid		Alanine	
	M	SD	M	SD	M	SD	M	SD	M	SD	M	SD	M	SD	M	SD	M	SD	M	SD	M	SD	M	SD	M	SD	M	SD	M	SD	M	SD
Cin_veru_01	9.57	0.05	0.8	0.04	10.43	0.07	3.95	0.03	0.09	0	0.7	0.01	9.16	0.04	6.44	0.04	3.07	0.11	4.74	0.02	3.94	0	0.06	0	0.55	0	1.95	0.02	3.4	0.17	1.6	0.02
Cin_veru_02	0.13	0	ND	ND	0.32	0	0.11	0	ND	ND	ND	ND	0.18	0	0.16	0	0.08	0	0.11	0	0.08	0	ND	ND	ND	ND	0.12	0	0.13	0.07	0.18	0
Cin_veru_03	1.14	0.01	0.17	0.01	0.8	0.05	0.3	0.02	ND	ND	0.12	0.01	0.58	0.03	0.7	0.04	0.21	0	0.35	0.02	0.24	0	ND	ND	ND	ND	0.25	0.01	ND	ND	0.22	0.01
Cin_cass_04	2.88	0.05	0.79	0.02	2.03	0.04	0.72	0.02	ND	ND	0.13	0	2.51	0.06	2.02	0.04	0.79	0.02	1.29	0.03	1	0	ND	ND	ND	ND	0.34	0.01	ND	ND	0.21	0.01
Cin_burm_05	1.61	0.05	0.58	0.01	1.79	0.03	0.62	0.01	ND	ND	0.12	0	2.03	0.03	1.54	0.02	0.75	0.02	1.1	0.02	0.83	0	ND	ND	0.06	0	0.35	0.01	ND	ND	0.27	0.01
Cin_cass_06	ND	ND	0.09	0.02	0.98	0.22	0.16	0.04	ND	ND	ND	ND	0.76	0.16	0.42	0.1	0.39	0.05	0.51	0.1	0.3	0.1	ND	ND	0.07	0	0.11	0.03	ND	ND	ND	ND
Cin_cass_07	ND	ND	ND	ND	1.79	0.07	0.05	0	ND	ND	0.16	0.01	0.34	0.02	0.09	0	1.93	0.21	0.15	0	0.14	0	ND	ND	0.14	0	ND	ND	ND	ND	ND	ND
Cin_veru_08	0.15	0.01	0.07	0.01	0.36	0.03	0.12	0.01	ND	ND	ND	ND	0.2	0.02	0.18	0.02	0.09	0.01	0.13	0.01	0.09	0	ND	ND	ND	ND	0.13	0.01	0.11	0.01	0.2	0.02
Cin_burm_09	0.21	0.02	0.15	0.01	0.48	0.05	0.25	0.02	ND	ND	ND	ND	0.61	0.07	0.51	0.06	0.23	0.01	0.33	0.03	0.23	0	ND	ND	0.31	0.1	0.1	0.02	0.48	0.06	0.06	0.01
Cin_veru_10	ND	ND	0.11	0	0.74	0.01	0.19	0.01	ND	ND	0.06	0	0.66	0.01	0.25	0	0.25	0.01	0.33	0.01	0.25	0	ND	ND	0.4	0	0.16	0.01	0.65	0.47	0.11	0
Cin_cass_11	ND	ND	1.13	0.02	32.64	0.38	2.59	0.04	ND	ND	1.81	0.03	27.46	0.36	3.59	0.05	11.46	0.07	14.89	0.16	10.67	0.1	ND	ND	1.84	0.1	1.46	0.02	0.09	0	0.54	0.01
Cin_cass_12	24.63	0.14	6.94	0.04	15.76	0.01	6.37	0.02	0.08	0	1.06	0.01	17.89	0.06	14.97	0.06	5.84	0.25	9.23	0.05	7.1	0	0.07	0	0.35	0	2.82	0.02	0.12	0	1.79	0.01
Cin_veru_13	1.92	0.23	0.3	0.02	16.52	0.09	4.33	0.05	0.88	0.01	1.43	0.03	6.95	0.04	3.53	0.04	2.34	0.09	3.81	0.02	3.05	0	0.09	0	0.44	0	4.7	0.05	0.35	0	6.09	0.05
Cin_veru_14	1.16	0.21	0.65	0.05	6.44	0.11	0.79	0.05	2.62	0.21	0.06	0	0.51	0.03	20.93	8.3	2.98	0.29	0.71	0.05	0.34	0.1	ND	ND	0.32	0	0.39	0.02	0.3	0.01	5.98	0.47
Cin_cass_15	23.86	0.64	2.6	0.09	10.86	0.29	3.87	0.11	ND	ND	4.55	0.11	10.23	0.35	6.7	0.19	4.33	0.38	4.86	0.19	3.55	0.1	ND	ND	0.39	0	2.76	0.09	1.03	0.11	4.06	0.12
Cin_veru_16	ND	ND	ND	ND	5.56	0.04	ND	ND	ND	ND	0.19	0	0.36	0.01	0.06	0	1.03	0.04	0.8	0.01	0.16	0	ND	ND	0.38	0	ND	ND	ND	ND	ND	ND
Cin_cass_17	ND	ND	0.15	0.01	4.73	0.36	0.35	0.03	ND	ND	0.25	0.02	3.82	0.29	0.49	0.04	1.62	0.15	2.12	0.14	1.54	0.1	ND	ND	0.16	0	0.18	0.02	0.14	0.06	0.07	0
Cin_cass_18	0.27	0.01	0.16	0	0.58	0	0.23	0	ND	ND	ND	ND	0.64	0	0.39	0	0.18	0	0.32	0	0.26	0	ND	ND	0.06	0	0.09	0	0.06	0	0.06	0
Cin_cass_19	0.3	0	0.16	0	0.56	0	0.23	0	ND	ND	ND	ND	0.6	0.01	0.45	0	0.17	0	0.3	0	0.24	0	ND	ND	0.06	0	0.1	0	0.07	0.01	0.07	0
Cin_cass_20	0.35	0	9.2	0.09	28.97	0.3	13.19	0.13	0.4	0.01	2.38	0.02	18.95	0.17	10.89	0.09	5.19	0.03	9.46	0.08	8.44	0.1	0.15	0	0.53	0	8.42	0.08	0.68	0.01	11.31	0.12
Cin_cass_21	1.97	0.03	3.72	0.01	12.04	0.03	3.76	0.02	0.13	0	0.89	0	6.6	0.02	2.68	0	2.38	0.04	3.59	0.02	2.82	0	0.13	0	0.52	0	2.14	0	0.17	0	2.59	0.01
Cin_veru_22	ND	ND	ND	ND	4.95	0.02	ND	ND	ND	ND	0.16	0	0.33	0	ND	ND	0.85	0.01	0.75	0.01	0.15	0	ND	ND	0.33	0	ND	ND	ND	ND	ND	ND
Cin_veru_23	ND	ND	ND	ND	5.04	0.05	ND	ND	ND	ND	0.16	0	0.33	0.01	0.06	0	0.91	0.04	0.77	0.01	0.15	0	ND	ND	0.33	0	0.05	0.01	0.09	0.01	ND	ND
Cin_veru_24	1.28	0.05	0.51	0.02	2.66	0.12	0.92	0.03	0.16	0.01	0.3	0.01	1.48	0.06	1.28	0.05	0.72	0.03	0.96	0.03	0.63	0	0.13	0	0.37	0	0.94	0.03	1.15	0.07	1.48	0.05
Cin_veru_25	5.23	0.12	0.5	0.11	20.81	0.88	4.96	0.15	0.14	0.01	2.63	0.07	19.13	0.78	6.79	0.16	6.78	0.18	8.91	0.36	7.5	0.3	0.3	0.01	0.95	0.1	4.4	0.12	1.44	0.05	3.23	0.1
Cin_veru_26	14.98	0.2	0.66	0.03	27.15	0.37	7.52	0.05	0.26	0	4.03	0.02	19.16	0.17	6.07	0.01	7.48	0.03	9.36	0.11	7.51	0.1	0.25	0	0.67	0	6.48	0.03	1.16	0.36	6.55	0.07
Cin_veru_27	2.99	0.12	0.97	0.07	13.58	0.42	3.72	0.12	0.77	0.02	1.54	0.05	6.1	0.19	3	0.09	1.88	0.06	3.28	0.11	2.7	0.1	0.07	0	0.43	0	4.06	0.11	0.31	0.01	5.35	0.12
Cin_veru_28	8.75	0.04	0.25	0.01	6.28	0.02	2.17	0.02	0.29	0	0.92	0.01	5.01	0.03	5.29	0.05	1.88	0.05	2.91	0.03	2.01	0	ND	ND	0.45	0	1.95	0.03	0.08	0	1.46	0.02
Cin_veru_29	0.35	0	0.58	0.05	61.79	1.21	42.99	1.15	0.4	0.01	6.39	0.17	18.65	0.4	2.59	0.06	12.1	0.4	7.75	0.14	9.34	0.2	0.28	0.01	0.78	0	5.57	0.15	0.55	0.27	1.48	0.05
Cin_burm_30	18.81	0.11	4.75	0.05	8.55	0.07	3.72	0.04	0.07	0	1.19	0	8.38	0.05	6.3	0.06	3.43	0.08	4.29	0.02	3.29	0	0.06	0	0.43	0	2.29	0.03	0.36	0.33	2.55	0.02
Cin_burm_31	21.99	0.2	5.63	0.07	11.42	0.15	4.67	0.06	0.08	0	0.83	0.01	12.3	0.15	4.7	0.12	4.42	0.08	6.47	0.06	5.01	0.1	0.07	0	0.41	0	2.12	0.03	0.43	0.14	1.73	0.02
Cin_burm_32	21.77	0.08	5.73	0.04	12.64	0.05	5.19	0.04	0.08	0	0.87	0	14.14	0.05	1.33	0.09	5.08	0.12	7.39	0.02	5.68	0	0.08	0	0.39	0	2.41	0.02	1.01	0.2	1.86	0.02
Cin_burm_33	22.57	0.15	5.87	0.03	12.34	0.18	4.89	0.03	0.08	0	0.86	0.01	13.34	0.18	1.03	0.06	4.41	0.08	6.99	0.06	5.43	0.1	0.08	0	0.41	0	2.19	0.02	0.48	0.27	1.76	0.01
Cin_burm_34	18.98	0.25	4.6	0	7.95	0.07	3.4	0.05	0.06	0	1.14	0	7.64	0.07	5.5	0.01	3.29	0.07	3.88	0.05	2.96	0	ND	ND	0.48	0	3.58	0.01	0.97	0.39	2.98	0.01
Cin_veru_35	ND	ND	ND	ND	3.17	0.09	ND	ND	ND	ND	0.12	0	0.3	0.01	ND	ND	0.67	0.01	0.53	0.02	0.13	0	ND	ND	0.36	0	0.06	0	1.56	1.32	0.06	0
Cin_veru_36	ND	ND	ND	ND	57.62	0.88	40.41	0.63	0.36	0	5.82	0.12	17.22	0.25	2.38	0.04	11.5	0.02	7.44	0.08	8.58	0.1	0.25	0	0.77	0	5.17	0.06	0.34	0.13	1.43	0.02
Cin_veru_37	13.47	1.23	1.12	0.09	3.2	0.24	1.92	0.14	0.17	0.01	0.72	0.07	3.86	0.3	6.68	0.51	1.46	0.06	2.34	0.18	1.44	0.1	ND	ND	0.29	0	2.11	0.13	0.32	0.19	1.63	0.09
Cin_veru_38	ND	ND	ND	ND	2.1	0.11	ND	ND	ND	ND	0.08	0	0.22	0	ND	ND	0.4	0.02	0.39	0.01	0.1	0	ND	ND	0.38	0	ND	ND	0.19	0.09	0.07	0.02
Cin_veru_39	0.31	0.01	0.49	0.17	65.4	0.51	36.83	0.34	0.35	0	6.21	0.05	19.86	0.14	4.69	0.06	13.86	0.24	9.34	0.06	9.63	0.1	0.24	0	0.8	0	5.49	0.06	0.78	0.06	1.41	0.01
Cin_cass_40	ND	ND	0.09	0.03	4.49	0.53	1.65	0.82	ND	ND	0.17	0.02	1.21	0.17	0.32	0.03	1.18	0.17	1.14	0.15	0.49	0.1	ND	ND	0.48	0	0.11	0.02	0.36	0.12	0.1	0.02
Cin_veru_41	ND	ND	ND	ND	2.01	0.04	ND	ND	ND	ND	0.08	0	0.21	0.01	ND	ND	0.43	0.01	0.35	0.01	0.09	0	ND	ND	0.32	0	ND	ND	0.69	0.03	0.06	0
Cin_veru_42	ND	ND	ND	ND	1.91	0.03	ND	ND	ND	ND	0.07	0	0.19	0.01	ND	ND	0.36	0	0.35	0	0.09	0	ND	ND	0.37	0	ND	ND	ND	ND	0.06	0
Cin_veru_43	ND	ND	ND	ND	1.91	0.02	ND	ND	ND	ND	0.07	0	0.19	0	ND	ND	0.36	0	0.34	0.01	0.09	0	ND	ND	0.35	0	ND	ND	ND	ND	ND	ND
Cin_veru_44	ND	ND	ND	ND	2.2	0.02	ND	ND	ND	ND	0.08	0	0.22	0	ND	ND	0.42	0	0.4	0.01	0.09	0	ND	ND	0.24	0	ND	ND	0.53	0.1	ND	ND
Cin_veru_45	9.61	0.14	0.21	0.02	6.04	0.12	2.19	0.04	0.23	0	0.9	0.02	5.7	0.08	5.37	0.07	2.05	0.05	3.14	0.04	2.22	0	ND	ND	0.41	0	2.07	0.03	0.4	0.02	1.57	0.02
Cin_veru_46	24.95	0.37	0.21	0.06	9.15	0.2	5.22	0.11	0.09	0	2.1	0.04	7.08	0.16	4.06	0.08	3.4	0.07	3.76	0.07	2.76	0.1	0.06	0	0.4	0	2.68	0.06	0.4	0.01	3.02	0.07
Cin_veru_47	ND	ND	0.38	0.09	68.01	0.6	38.28	0.57	0.36	0	6.57	0.06	21.08	0.18	5.12	0.04	13.94	0.19	9.82	0.05	10.19	0.1	0.25	0	0.83	0	5.71	0.06	0.51	0.21	1.47	0.02
Cin_veru_48	ND	ND	ND	ND	4.71	0.07	ND	ND	ND	ND	0.16	0	0.31	0.01	ND	ND	0.84	0.02	0.68	0.01	0.14	0	ND	ND	0.36	0	ND	ND	0.52	0.18	0.09	0

## Data Availability

The datasets generated for this study can be found in the Appendix A, including the 1H-NMR spectra as normalized binned buckets.

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
