# Peer review of "Nuclear Magnetic Resonance Fingerprints and Mini DNA Markers for the Authentication of Cinnamon Species Ingredients Used in Food and Natural Health Products"

_plants, 2024, doi:10.3390/plants13060841_

Round 1

Reviewer 1 Report

Comments and Suggestions for Authors

The presented manuscript NMR fingerprints and mini-DNA markers for authentication of cinnamon species ingredients used in food and natural health products is focused on the identification of true cinnamon. Cinnamon, although not the most important crop used for human nutrition, is a widely used and valued spice worldwide. Therefore, the issue of authentication is of importance and I consider the study to be useful and timely. 

The authors focused on the authentication of cinnamon using mini-DNA markers and NMR fingerprints. This provided more relevant and accurate data than has been published to date. 

Although I rate the work very positively, I still have definite uncertainties in the manuscript:

1) For DNA extraction, you state that a commercially available kit was used. According to Tables S1 and S2, different material was used. Were there any differences between the different types of input material? Which is the most suitable for the extraction. In my experience with this kit, the DNA is not stable enough and degrades quite quickly. 

2) Consider whether it would be useful to insert table S4 in the text, or at least list the Exon/intron regions for clarity. These data are not adequately described in the text. 

3) Figure 1 is very interesting, so it would be useful to change the quality of the figure, especially the figure legend. 

Author Response

RESPONSES TO REVIEWER

We appreciate all the constructive comments made by the reviewer, which have helped to improve the manuscript.

Reviewer 1:

1) For DNA extraction, you state that a commercially available kit was used. According to Tables S1 and S2, different material was used. Were there any differences between the different types of input material? Which is the most suitable for the extraction. In my experience with this kit, the DNA is not stable enough and degrades quite quickly. 

Author Response: edits completed. We added the following text to the discussion on lines 275-277 “In our study we found that DNA quantity was higher in fresh leaves and lower in processed extract powders, which is supported by earlier published research [59,60].”

2) Consider whether it would be useful to insert table S4 in the text, or at least list the Exon/intron regions for clarity. These data are not adequately described in the text. 

Author Response: edits completed. We moved Table S4 to the text as Table 1.

3) Figure 1 is very interesting, so it would be useful to change the quality of the figure, especially the figure legend. 

Author Response: edits completed. We improved the quality of Figure 1 including the figure legend.

Reviewer 2 Report

Comments and Suggestions for Authors

Dear authors, I have read the paper under title „NMR fingerprints and mini-DNA markers for authentication of cinnamon species ingredients used in food and natural health products“, written by Subramanyam Ragupathy et al.

It is an interesting topic and very important for the food and spice industry. But, I think that some things should be corrected:

Abstract

Line 11 – it should be „burmannii (toxic coumarin)“.....

Introduction

Line 39 – authors are using  `cassia` cinnamon. If this is the name, it should be with the capital letter, or with Latin name. Correct this here and throughout the whole text.

Line 61 – Which cinnamon species?

Line 62/63 – Sentence that starts with `These methods are very…..` should be deleted, it is not necessary, and not scientifically important, because all modern lab methods require skilled technicians.

Lines 70/78 – Make it shorter…. NO need to talk much about NMR…

Lines 88/95 – Like the previous comment. Make it condensed.

Lines 110/117 – The same….too thorough   And many more lines…..   2,5 pages of the Introduction should be cut down to maximum 1,5 pages.
The text is too thorough, many unnecessary things are described.
This a scientific paper, not a review.
  Material and methods Honestly, I have no clue how many samples were used.
First it was written that it was ` 22 samples of 3 species were gathered`,
then ` The thirteen leaf samples of all species were identified`….and later for
NMR analysis it was written `48 samples of 3 species were gathered from the NHPRA voucher…`
I suggest authors to explain it better, and refer the text with supplementary material.

Discussion

Too general, too long. No results and no story about 16 bioactive molecules you have detected with NMR method. Can you quantify it? Can you compare the quantity between the species? This should be very important. Why didn`t you determine coumarin, this is the most important compound!!! This should be the core of the paper. Rewrite it.

Conclusion

Too general. No real conclusions are withdrawn, only that we can use NMR and mini DNA markers to distinguish cinnamon species. The question is, what should be looked for to make the differences, both in mini DNA and chloroplasts genomes. And what about NMR? What are the most important compounds? What compounds are making the differences between the species? Rewrite it.

Author Response

RESPONSES TO REVIEWER

We appreciate all the constructive comments made by the reviewer, which have helped to improve the manuscript.

Reviewer 2:

It is an interesting topic and very important for the food and spice industry. But, I think that some things should be corrected:

Abstract

Line 11 – it should be „burmannii (toxic coumarin)“.....

Author Response: edits completed

Introduction

Line 39 – authors are using  `cassia` cinnamon. If this is the name, it should be with the capital letter, or with Latin name. Correct this here and throughout the whole text.

Author Response: edits completed throughout the text

Line 61 – Which cinnamon species?

Author Response: edits completed. We added the other species names (C. burmannii, C. cassia and C. loureiroi) according to the cited article.

Line 62/63 – Sentence that starts with `These methods are very…..` should be deleted, it is not necessary, and not scientifically important, because all modern lab methods require skilled technicians.

Author Response: edits completed. We deleted the sentence ’These methods are very robust and sensitive but are costly and require highly skilled technicians.’

Lines 70/78 – Make it shorter…. NO need to talk much about NMR…

Author Response: edits completed. We shortened the section on NMR keeping some relevant background information on NMR.

Lines 88/95 – Like the previous comment. Make it condensed.

Author Response: edits completed. We shortened the section on NMR keeping some relevant background information on the limitations of previous studies.

Lines 110/117 – The same….too thorough   And many more lines…..   2,5 pages of the Introduction should be cut down to maximum 1,5 pages. The text is too thorough, many unnecessary things are described. This a scientific paper, not a review.  

Author Response: edits completed. We shortened the introduction by a half page. We feel that the remaining text is needed to provide the proper background for the study to a broader scope of readers and industry members.

Material and methods Honestly, I have no clue how many samples were used.

First it was written that it was ` 22 samples of 3 species were gathered`,

then ` The thirteen leaf samples of all species were identified`….and later for

NMR analysis it was written `48 samples of 3 species were gathered from the NHPRA voucher…`

I suggest authors to explain it better, and refer the text with supplementary material. 

Author Response: edits completed. We edited for clarification of sample numbers used in the methods within section 4.1 and 4.3:

4.1. DNA Samples Extraction & Sequencing

22 samples of 3 species were gathered including leaf and bark samples (15) and industry powdered samples of known prevenance (7) from the Natural Health Products Research Alliance (NHPRA) collections, College of Biological Sciences, University of Guelph, Canada (Supplemental Table S1).

4.3. Nuclear Magnetic Resonance (NMR) sample preparation methods

48 samples of 3 species were used from the NHPRA collections, including 7 powdered samples used for DNA analysis in this study (Supplemental Table S2); note that the bark and leaf samples were not used in the NMR study as they have different chemical profiles than powdered samples whereas the DNA in all sample matrices contains the same DNA sequence profiles.

Discussion

Too general, too long. No results and no story about 16 bioactive molecules you have detected with NMR method. Can you quantify it? Can you compare the quantity between the species? This should be very important. Why didn`t you determine coumarin, this is the most important compound!!! This should be the core of the paper. Rewrite it.

Author Response: Yes, we agree this is important and a gap in research literature. The methods developed in the current manuscript were focused on authentication of cinnamon species ingredient verification as a separate R&D project. We have another R&D project in which we are developing quantitative methods using spiking experiment with pure bioactive compounds. This will provide a method to assess the quantity between different species, and product formulations. This manuscript will be developed in 2024 and submitted as a separate publication in order to keep a clear focus on the different goals of these research projects.

Conclusion

Too general. No real conclusions are withdrawn, only that we can use NMR and mini DNA markers to distinguish cinnamon species. The question is, what should be looked for to make the differences, both in mini DNA and chloroplasts genomes. And what about NMR? What are the most important compounds? What compounds are making the differences between the species? Rewrite it.

Author Response: We agree with the reviewers ideas and are working on these goals in another project – please see comment above. The methods developed in the current manuscript were focused on authentication of cinnamon species ingredient verification as a separate R&D project. Our research addressed this gap in the literature and provides novel tools for quality assessment of cinnamon ingredients. We will be publishing quantitative R&D that provides methods and analysis of bioactive compounds among important cinnamon species.

Round 2

Reviewer 2 Report

Comments and Suggestions for Authors

Dear authors,

Small make up has been made in the paper, but the core is still the same. No editing in the `Discssion` and `Conclusions` section.

Even if you are planning to do quantities methods in the next paper you should explain them here. Why did you do NMR if you don`t want to talk about the results. If you don`t want to discuss about these results, exclude this method from the paper.

The same is for the `Conclusion`, nothing about mini DNA and chloroplasts genomes and NMR.

I suggest that you should do this other study you are planning and then write one complete paper.

Author Response

Small make up has been made in the paper, but the core is still the same. No editing in the `Discssion` and `Conclusions` section.

Thank you for your comment; We have decided to include the quantification analysis as you suggested which has taken some time to complete.  We have included in the manuscript the quantities of the 16 metabolites and discussed the observed results in the manuscript.

Even if you are planning to do quantities methods in the next paper you should explain them here. Why did you do NMR if you don`t want to talk about the results. If you don`t want to discuss about these results, exclude this method from the paper.

We have incorporated quantification methods, and results and discussed our observations in their respective sections.

The same is for the `Conclusion`, nothing about mini DNA and chloroplasts genomes and NMR.

We provided a summary based on the goals of this manuscript. Based on the results, the mini-DNA primers can be a quick tool to access the species of the product. NMR fingerprints provide a tool for species identity and detailed information concerning the variation in quantity of molecules for each of the cinnamon species.

I suggest that you should do this other study you are planning and then write one complete paper.

As your suggestion, we have included the NMR quantification results in this manuscript.

Round 3

Reviewer 2 Report

Comments and Suggestions for Authors

Dear authors,

the manuscript is much improved. I am glad that you have accepted my suggestions.

It is much better now.